# Variability in Anthocyanin Expression in Native Maize: Purple *Totomoxtle* as a Phenotypic Trait of Agroecological Value

**DOI:** 10.3390/plants14162511

**Published:** 2025-08-12

**Authors:** José Gregorio Joya-Dávila, Federico Antonio Gutiérrez-Miceli, Leslie Alondra Serrano-Gómez, Alexis Salazar-Navarro, Daniel González-Mendoza, Olivia Tzintzun-Camacho, Ana Martín Santos-Espinoza, Grisel Sánchez-Grajalez, Eraclio Gómez-Padilla, Jaime Llaven-Martínez

**Affiliations:** 1Laboratorio de Biotecnología Instituto de Ciencias Agrícolas, Universidad Autónoma de Baja California, Carretera a Delta, Ejido Nuevo León s/n, Mexicali 21705, Baja California, Mexico; alexis.salazar@uabc.edu.mx (A.S.-N.);; 2División de Posgrado, Instituto Tecnológico de Tuxtla-Gutiérrez, Carretera Panamericana km 1080, Tuxtla Gutiérrez 29050, Chiapas, Mexico; leslie.serranog23@gmail.com (L.A.S.-G.);; 3Facultad de Ciencias Agronómicas, Campus V., Universidad Autónoma de Chiapas (UNACH), Villaflores 30470, Chiapas, Mexico

**Keywords:** native maize landraces, pigment-based selection, anthocyanin accumulation, nutritional metabolites, *totomoxtle* pigmentation, phenotypic trait conservation

## Abstract

Purple *totomoxtle* (maize husk) in native maize represents a phenotypic trait of cultural and agronomic significance within traditional Mesoamerican agroecosystems. This study evaluated the phenotypic expression of anthocyanins in vegetative and reproductive tissues of ten native maize genotypes, including inter-parental crosses derived from both pigmented and non-pigmented lines. Field trials were conducted under rainfed conditions in Chiapas, Mexico. Visual and quantitative assessments included pigmentation intensity, chlorophyll and carotenoid content, ear traits and appearance, grain health, and yield performance. Genotypes exhibiting the purple phenotype showed consistent pigment accumulation in stems, nodes, leaf sheaths, tassels, and bracts (totomoxtle), with statistically significant differences compared to non-pigmented controls. Anthocyanin content in totomoxtle increased by 30% during late developmental stages, whereas chlorophyll and carotenoid levels peaked during early vegetative growth. Pigmented materials displayed healthier grain, enhanced ear appearance, and higher yields, with the JCTM × LLMJ cross reaching 6.60 t ha^−1^. These findings highlight the functional value of purple totomoxtle and its potential in agroecological programs aimed at resilience, genetic conservation, and integral resource utilization, providing useful criteria such as stable pigment expression and superior yield to guide sustainable reproduction strategies.

## 1. Introduction

Maize (*Zea mays* L.), a member of the Poaceae family, is a Mesoamerican crop domesticated over 9000 years ago, with its center of origin located in regions that now comprise Mexico and Guatemala [1,2]. It is one of the fundamental pillars of global agri-food systems, owing to its nutritional value and its versatility as a raw material in the food, energy, and pharmaceutical industries. In Mexico, national average yields are estimated at 3.9 tons per hectare [3,4], with native landraces primarily cultivated under agroecological conditions and used largely for local consumption.

The conservation of these varieties not only represents biocultural heritage but also a strategic source of desirable traits such as *totomoxtle* pigmentation, which can be incorporated into breeding programs to increase productivity without compromising genetic identity.

Globally, maize production surpassed 1.2 billion tons in 2023, consolidating its role as a strategic component of the world’s food systems [5]. In Mexico, maize is not only a staple food but also a symbol of cultural identity and agricultural biodiversity, with over 60 native landraces preserved in traditional farming systems [6]. As the center of origin and genetic diversification of maize, Mexico plays a key role in conserving germplasm adapted to diverse environmental conditions [7]. This genetic diversity is critical for addressing challenges associated with climate change, including water, heat, and salinity stress, which threaten yield stability and food security in rural communities [8].

Among this diversity, the purple pigmentation of the totomoxtle stands out as a phenotypic trait associated with anthocyanin accumulation, bioactive compounds of functional and cultural interest [9,10]. In this sense, the study of the physiological and adaptive responses of corn to stress conditions is essential for the generation of resilient cultivars and the consolidation of sustainable production systems.

Although all cultivated maize belongs to the same species (*Zea mays* L.), it exhibits remarkable morphological diversity in terms of plant architecture, floral structures, ears, and kernels. This phenotypic variation allows for classification and differentiation of races, varieties, and hybrids [4,11].

In recent years, scientific interest has expanded toward bioactive compounds in maize vegetative structures beyond the grain, such as the bracts or totomoxtle, which contain high levels of anthocyanins and carotenoids—compounds with well-recognized antioxidant and nutraceutical potential [9,12].

This has led to new lines of research not only aimed at improving kernel nutritional quality but also at identifying functional compounds in other plant tissues—such as bracts, stems, and leaves—to promote whole-plant utilization and reduce postharvest losses [13,14].

Maize color is directly related to its phytochemical content and associated health benefits. These bioactive compounds are produced by plants as defense mechanisms, and thus are present in most maize types. Blue, purple, black, pink, and red maize varieties contain anthocyanins—pigments responsible for their coloration and recognized for their antioxidant activity [15]. Yellow and orange maize varieties contain carotenoids, precursors of vitamin A and associated with visual health [16]. Additionally, red and pink maize contain flavonoids, and many types also accumulate phenolic acids, all of which can reduce cellular oxidation caused by free radicals [4,17].

One vegetative structure that has gained increasing attention is the totomoxtle, the common name for the bracts that surround the maize ear during development. When dry, these husks have traditionally been discarded or used in Mexico for wrapping tamales and in artisanal crafts [18].

Although historically undervalued in agro-industrial contexts and underexplored as a source of bioactive compounds, totomoxtle is now emerging as a tissue rich in secondary metabolites, particularly anthocyanins and carotenoids. In certain native genotypes, it exhibits intense purple pigmentation—a distinctive phenotypic trait with antioxidant properties and nutraceutical potential, as reported in recent studies [17,19,20].

Beyond its functional value, totomoxtle holds strong cultural significance. Its use in traditional Mexican cuisine not only reflects culinary heritage but also influences the sensory properties of food, as pigments can transfer to dough, imparting color and distinctive flavor [20].

Furthermore, totomoxtle pigmentation may be associated with genetic traits linked to tolerance against biotic and abiotic stress, making it a valuable structure for understanding defense mechanisms under adverse conditions [21]. The content and profile of these compounds vary widely among landraces, local varieties, and improved materials, underscoring the importance of phytochemical characterization [22].

Studying totomoxtle contributes not only to the conservation of native germplasm but also to the revalorization of a traditionally marginalized by-product, in alignment with sustainable production strategies that integrate health, culture, and biodiversity.

Nevertheless, the limited scientific exploration of totomoxtle has hindered its full utilization in industries such as food, cosmetics, and pharmaceuticals, as well as its inclusion in genetic improvement programs. Since anthocyanin and carotenoid concentrations are influenced by genotype, environmental conditions, and agronomic practices, it is a priority to identify races and hybrids with high levels of functional pigments. Such knowledge is essential for promoting agro-industrial value chains, supporting circular economy strategies, and conserving native varieties through place-based approaches.

In this context, purple pigmentation in totomoxtle can serve as a practical visual indicator for agroecological selection, as it is associated with high anthocyanin accumulation.

Accordingly, there is a growing need to develop improved native maize varieties that retain traditional traits such as purple totomoxtle (linked to high anthocyanin content), while also offering increased yield and tolerance to adverse conditions [19]. This vision responds to the urgent need to enhance food security, preserve genetic biodiversity, and increase the added value of native maize within a transforming agri-food system.

Research by Llaven et al. [23], focused on the phenological and morphological characterization of local maize from Villaflores, Chiapas, has identified parental materials expressing purple totomoxtle, laying the groundwork for studies such as the present one.

The aim of this study was to identify experimental hybrids with high agronomic potential and functional phytochemical profiles, based on crosses between pigmented native maize genotypes from southeastern Mexico. We focused on analyzing the phenotypic expression of purple totomoxtle and its relationship with pigment accumulation and field performance. The characterization of these materials seeks to support the selection of new cultivars that integrate productivity, resilience, and nutraceutical value—promoting a more sustainable and holistic use of native maize in key regions such as Chiapas.

## 2. Results

### 2.1. Anthocyanins in Vegetative and Reproductive Structures

The native maize genotypes evaluated in this study exhibited notable genetic variability associated with anthocyanin pigmentation in both vegetative and reproductive tissues. All genotypes selected for the conservation of the purple totomoxtle phenotype showed visible and consistent anthocyanin expression across multiple structures (Figure 1). At the vegetative level, pigmentation was clearly observed in stems and leaf sheaths, particularly in the basal region of the plants, as well as in primary and adventitious roots, as illustrated in the corresponding images. This coloration was also prominent in the basal nodes, indicating localized expression of phenolic compounds from early developmental stages.

In reproductive structures, anthocyanin presence was evident in the male inflorescences, specifically in the anthers and entire spikelets during anthesis. In the ear, pigmentation was observed in the silks (stigmas), and cross-sectional analysis confirmed the presence of purple coloration in the inner bracts (totomoxtle) and rachis (cob). In contrast, the kernels remained predominantly white, supporting the hypothesis that anthocyanin accumulation occurs in a structured and tissue-specific manner. Non-selected genotypes (used as controls) showed no detectable pigmentation in any of the evaluated structures.

### 2.2. Phenotypic Expression of Anthocyanins in Totomoxtle and Nodal Structures

Figure 2 displays the differential anthocyanin pigmentation in two key structures across ten native maize genotypes (Figure 2A–J): totomoxtle (Section 1) and the basal region of the stem, including nodes and leaf sheaths (Section 2).

Section 1—Pigmentation in totomoxtle (bracts): In the upper section of Figure 2, transverse ear sections reveal intense bract pigmentation (totomoxtle) in genotypes MCTM# (Figure 2A), LLMJ# (Figure 2B), JCTM# (Figure 2C), LLMJ × JCTM (Figure 2D), and JCTM × LLMJ (Figure 2E). All these materials derive from pigmented lines or directed crosses and exhibit purple pigmentation with variations in tone and concentric distribution.

The genotype JCTM × RP Rojo (Figure 2F), with a pigmented female parent (JCTM#) and a non-pigmented male parent (Rojo Parral), showed intermediate expression of the trait. In contrast, genotypes Mor × MC (Figure 2G), Mor# (Figure 2H), MC × Mor (Figure 2I), and MC# (Figure 2J) did not exhibit visible bract pigmentation, confirming their role as non-selected controls for anthocyanin expression.

Section 2—Pigmentation in stems, nodes, and basal sheaths: The lower section of Figure 2 documents anthocyanin presence in nodes and leaf sheaths at 43 days after sowing (DAS). The genotypes with the most intense pigmentation were LLMJ# (Figure 2B), LLMJ × JCTM# (Figure 2D), JCTM × RP Rojo (Figure 2F), and JCTM# (Figure 2C), all displaying vivid coloration in basal sheaths. A second group with moderate pigmentation, restricted to the first few internodes, included JCTM# × LLMJ# (Figure 2E) and MCTM# (Figure 2A).

Although lacking totomoxtle pigmentation, genotypes Mor# (Figure 2H) and MC × Mor (Figure 2I) exhibited anthocyanin accumulation in the first three basal nodes, indicating localized trait activation. In contrast, Mor × MC (Figure 2G) and MC# (Figure 2J) showed no visible pigmentation in these structures.

Native maize genotypes exhibited marked differences in anthocyanin expression at basal stem nodes, with variation in both the number of pigmented nodes and pigment intensity (Table 1). The highest values were recorded in genotypes LLMJ × JCTM#, LLMJ#, and JCTM × LLMJ#, showing significant differences (*p* ≤ 0.05), which reflects consistent trait expression in selected lines with pigmented backgrounds. Notably, LLMJ#, previously selected for totomoxtle pigmentation, showed stronger expression when used as the male parent (JCTM × LLMJ) compared to its reciprocal cross (LLMJ × JCTM).

In contrast, the JCTM × RP Rojo cross, which combined a pigmented parent with one lacking bract pigmentation, displayed a significant reduction in both the number and intensity of pigmented nodes. This result highlights the influence of the non-pigmented genetic background on trait modulation. The genotypes classified as negative phenotypic controls for totomoxtle pigmentation (Mor × MC, Mor#, MC × Mor, and MC#) exhibited low to absent nodal pigmentation, confirming their profile as reference materials lacking trait expression.

### 2.3. Population-Level Classification of Anthocyanin Intensity in Vegetative Structures

At 65 days after sowing (DAS), a general visual assessment of anthocyanin pigmentation in whole vegetative structures—mainly stems and leaf sheaths—allowed classification of each genotype’s plant population into three categories: high, intermediate, and absent pigmentation (Figure 3). This analysis complemented previous node-specific evaluations and enabled estimation of the overall proportion of pigmented plants per genotype under field conditions.

Genotypes LLMJ#, LLMJ × JCTM, JCTM × LLMJ, and JCTM# stood out, with the highest proportions of plants exhibiting intense pigmentation, reaching 72%, 56%, 56%, and 60%, respectively. These results are consistent with their phenotypic selection history for pigmented totomoxtle, reflecting a generalized expression of the trait throughout the plant.

The genotype JCTM × RP Rojo showed an intermediate pattern, with 44% of plants classified under the high pigmentation category, while MCTM# recorded 36%, indicating a degree of intra-genotypic variability in phenotypic expression.

The intermediate pigmentation category showed similar percentages across most genotypes, with no statistically significant differences, which may suggest transitional expression influenced by environmental conditions or partial genetic interactions.

In the absence category, genotypes Mor × MC, Mor#, MC#, and MC × Mor showed the highest proportions of non-pigmented plants (100%, 68%, 60%, and 68%, respectively). A noteworthy observation was made in the JCTM × RP Rojo cross, where 36% of the plants lacked visible pigmentation despite the pigmented background of the female parent (JCTM#).

### 2.4. Anthocyanin and Photosynthetic Pigment Accumulation

The quantification of total anthocyanin content in leaf sheaths and totomoxtle showed patterns consistent with previously observed visual expression (Table 2). In the sheath tissue, genotype LLMJ# exhibited the highest value (142.3 mg L^−1^), significantly greater than all others (*p* < 0.0001), confirming its potential as a genetic source of pigmentation. It was followed by genotypes JCTM# (100.8 mg L^−1^) and JCTM × LLMJ (94.5 mg L^−1^), with no statistical difference between them. In the reciprocal crosses, higher anthocyanin content was observed when JCTM# served as the female parent, suggesting a possible influence of parental origin on the expression of the pigmentation trait.

Interestingly, the genotype MC × Mor showed a higher anthocyanin content than either of its individual parents (65.7 mg L^−1^), possibly due to favorable non-additive genetic interactions enhancing pigment synthesis. Conversely, the cross JCTM × RP Rojo—which includes a non-pigmented parental line—exhibited a statistically significant reduction in anthocyanin content (59.6 mg L^−1^), reinforcing the influence of the non-pigmented parent on this trait’s suppression.

Regarding the totomoxtle, significant differences were found among genotypes (*p* = 0.0102), with LLMJ# showing the highest concentration (376.9 mg L^−1^), followed by JCTM#, JCTM × LLMJ, and LLMJ × JCTM. These four genotypes formed a statistically superior group compared to the rest of the treatments, which included genotypes with lower phenotypic expression such as MCTM#, JCTM × RP, MC × Mor, Mor#, MC#, and Mor × MC.

In relation to the phenological sampling stage, no significant differences in sheath anthocyanin content were observed between days 19, 43, and 65, indicating stable pigment accumulation during early vegetative development. In contrast, significant differences were detected in totomoxtle, with a higher concentration at day 135 (123.6 mg L^−1^) compared to day 72 (95.3 mg L^−1^), suggesting that anthocyanin accumulation in this structure increases with physiological maturity.

### 2.5. Dynamics of Photosynthetic Pigments

The concentrations of chlorophyll a, chlorophyll b, total chlorophyll, and carotenoids showed significant variation among the evaluated native maize genotypes and their sibling crosses, as well as across sampling days (Appendix A). These pigments serve as key indicators of the functional status of the photosynthetic apparatus during the establishment and leaf expansion phases.

In general, genotypes not selected for anthocyanin expression in totomoxtle exhibited the highest chlorophyll content. However, some notable exceptions were observed among pigmented genotypes. The cross LLMJ × JCTM recorded the highest chlorophyll a concentration among pigmented materials (50.45 mg mL^−1^), with statistically significant differences compared to its parents. Similarly, the cross JCTM × RP Rojo, which combines a pigmented parent with a non-pigmented one, showed an increase in chlorophyll a content (41.72 mg mL^−1^) relative to its female parent JCTM# (26.96 mg mL^−1^), suggesting a positive effect of the non-pigmented male parent on this physiological trait.

For chlorophyll a, the genotype MC × Mor stood out with the highest concentration (56.37 mg mL^−1^), followed by MC# (50.99 mg mL^−1^), LLMJ × JCTM (50.45 mg mL^−1^), and Mor# (45.75 mg mL^−1^), all grouped statistically. A similar pattern was observed for chlorophyll b, with MC × Mor, MC#, and LLMJ × JCTM showing the highest concentrations (21.41, 19.17, and 18.63 mg mL^−1^, respectively). In contrast, pigmented genotypes such as LLMJ#, JCTM#, and MCTM# showed the lowest values, all below 11 mg mL^−1^.

The trend for total chlorophyll followed a similar pattern: genotypes without totomoxtle pigmentation had the highest values, while among pigmented materials, only LLMJ × JCTM approached those levels, exceeding 69 mg mL^−1^. This positions the cross as one of the most physiologically efficient among pigmented genotypes.

Regarding carotenoid content, a slightly different pattern was observed. The highest value was recorded in Mor × MC (22.83 mg mL^−1^), followed closely by MC × Mor, LLMJ × JCTM, and MC#, all of which exceeded 21 mg mL^−1^. In contrast, the selected pigmented genotypes (LLMJ#, JCTM#, MCTM#) displayed the lowest carotenoid levels, with concentrations below 12 mg mL^−1^. These results reinforce the trend of reduced photosynthetic pigment expression in genotypes with high anthocyanin content.

From a phenological perspective, a clear dynamic was observed in pigment accumulation. The highest concentrations of chlorophyll a, b, and total chlorophyll were recorded at day 43, during the active leaf expansion stage (65.63, 20.28, and 85.91 mg mL^−1^, respectively), representing the optimal physiological point for assessing photosynthetic efficiency in these materials.

In contrast, carotenoids reached their peak concentration at day 19 (22.67 mg mL^−1^), then gradually decreased through day 65 (8.24 mg mL^−1^). This pattern suggests a more active role for these antioxidant compounds during early development stages, possibly linked to protection of the photosynthetic apparatus under initial stress conditions.

### 2.6. Ear Quality and Grain Health

The evaluation of phenotypic traits related to agronomic ear quality revealed slight variability among genotypes, particularly in overall ear appearance, where statistically significant differences were observed (Table 3).

Genotypes JCTM# (8.70) and LLMJ# (8.67) achieved the highest scores for ear appearance, followed by JCTM × RP (8.50) and MC × Mor (8.47). These materials exhibited compact ears with full husk coverage and morphological uniformity within plots—traits aligned with the selection criteria established during the initial phases of the breeding program.

In contrast, genotype Mor# showed the lowest score (8.07), falling within the lower statistical group along with Mor × MC (8.33). Although the absolute differences were small, they reflect reduced structural uniformity and possible morphological variation within these populations.

Regarding damaged grain biomass and overall grain health, no significant differences were found among treatments. However, damaged grain biomass was minimal across all genotypes, with an overall average of 0.252 g per plot, indicating favorable field conditions and low postharvest incidence of pathogens or pests.

The lowest values were recorded in MCTM# (0.089 g), JCTM# (0.149 g), and MC × Mor (0.177 g), suggesting a consistent relationship between good external appearance and low internal grain damage. In terms of grain health, most genotypes were classified between 1.00 and 1.33, corresponding to healthy kernels or those with minimal damage. Only Mor × MC reached a value of 1.67 (although not statistically different); this may indicate a slight susceptibility in this non-pigmented material.

### 2.7. Grain Yield

Grain yield analysis across the ten evaluated genotypes showed an overall average of 5.46 t ha^−1^, with substantial variability attributed to genetic background and type of cross (Figure 4). The highest yielding genotype was the JCTM × LLMJ cross, which reached 6.60 t ha^−1^, significantly exceeding the trial average.

This cross outperformed its male parent JCTM# (6.13 t ha^−1^) by 0.47 t ha^−1^ and its female parent LLMJ# (5.68 t ha^−1^) by 0.92 t ha^−1^, demonstrating a positive effect from combining both pigmented lines. Other genotypes with yields above the overall average included LLMJ × JCTM (6.00 t ha^−1^), MCTM# (5.58 t ha^−1^), and JCTM# (6.13 t ha^−1^), all of which expressed pigmentation in the totomoxtle.

In contrast, genotypes without prior selection for anthocyanin traits exhibited the lowest yields, with Mor × MC (4.66 t ha^−1^) recording the lowest productivity, falling below the general average.

## 3. Discussion

### 3.1. Importance of Conserving the Pigmented Phenotype in Native Maize

The native maize genotypes evaluated in this study exhibited remarkable genetic variability in anthocyanin expression across both vegetative and reproductive structures. This diversity highlights the richness of local germplasm and supports the value of participatory conservation strategies implemented in the Frailesca region of Chiapas since the year 2000. In genotypes selected for their purple totomoxtle, pigmentation was expressed consistently and broadly in stems, sheaths, male inflorescences, roots, and bracts, suggesting coordinated phenotypic expression likely regulated by multiple loci involved in the anthocyanin biosynthesis pathway [24,25].

In reproductive structures, pigmentation observed in anthers, spikelets, and stigmas during anthesis may serve a protective function, shielding genetic material from UV radiation or heat stress, thereby helping to maintain pollen viability and fertility. Additionally, the persistence of pigment in the totomoxtle and rachis may be associated with defense mechanisms that protect the ear and kernels during grain filling and maturation [11].

It has been demonstrated that anthocyanin accumulation in organs such as roots, stems, and leaves contributes to greater tolerance to water, heat, and biotic stress due to its antioxidant properties and its role in scavenging reactive oxygen species [26].

Moreover, recent transcriptomic studies in maize have identified coordinated gene pathways linking anthocyanin biosynthesis with antioxidant mechanisms induced by abiotic stress, particularly drought and heavy metal toxicity [27]. This evidence suggests that anthocyanin accumulation is not only compatible with photosynthetic performance but may also actively contribute to physiological resilience under adverse environmental conditions.

Recent reviews confirm that anthocyanins act as “natural sunscreens,” absorbing green-blue-UV light and potentially reducing photooxidative damage in chloroplasts. Furthermore, they function as metal chelators and modulators of defense-related metabolic pathways under abiotic stress conditions [28]. In addition, recent functional analyses have identified the role of transcription factors such as *MYB75* and *bHLH122*, which not only regulate anthocyanin biosynthesis but also activate antioxidant and photoprotective pathways, helping to maintain homeostasis under abiotic stress conditions [29]. This genetic and physiological integration supports the idea that pigmented genotypes may offer a real adaptive advantage, especially in environments with high radiation or extreme temperatures.

In addition, recent studies using omics technologies have shown that pigmented maize genotypes possess greater metabolic diversity in pigments and functional compounds, reinforcing their value as functional foods and raw materials for natural dye extraction [28,29]. This added value strengthens the case for preserving phenotypically diverse crops and highlights the importance of community-based cultivation systems that maintain their genetic stability and functional diversity [26,30].

In this study, pigmented genotypes derived from LLMJ# and JCTM# exhibited a stable systemic distribution of pigmentation from 43 days after sowing through the reproductive stage, suggesting that this trait can be selected and maintained in the field as an adaptive marker. Although molecular markers were not analyzed, the patterns observed in reciprocal crosses suggest complex non-additive inheritance mechanisms, possibly related to cytoplasmic gene regulation or epigenetic effects, in agreement with previous findings in pigmented maize [24,31].

The phenotypic assessment at 65 days after sowing confirmed these patterns. Classification of plants by visible pigmentation intensity revealed a high proportion of individuals with maximum pigmentation among selected genotypes (up to 72% in LLMJ#), indicating stable and systemic expression. In contrast, non-selected genotypes functioned as negative controls, lacking significant anthocyanin accumulation. This contrast underscores the influence of genetic background and selection schemes and suggests likely additive or epistatic inheritance in crosses such as LLMJ × JCTM and JCTM × LLMJ [32,33].

### 3.2. Agronomic Quality and Yield

The results obtained for agronomic quality components revealed a clear advantage in pigmented genotypes, particularly those involved in directed crosses. The parental lines JCTM# and LLMJ# stood out with the highest scores for ear appearance, which was associated with greater structural uniformity, good husk coverage, and reduced incidence of visible damage in the field. These features reflect a phenotypic selection history that prioritized morpho-functional traits of both agronomic and cultural value in traditional contexts.

In terms of grain health, damaged grain biomass remained low across all treatments, though it was even lower in pigmented genotypes. Notable values were observed in MCTM# (0.089 g), JCTM# (0.149 g), and MC × Mor (0.177 g), all showing minimal damage levels. While differences were not statistically significant, the trend suggests a possible association between anthocyanin expression and enhanced resistance to pathogens or postharvest deterioration. Anthocyanins are known for their antioxidant activity and documented antimicrobial effects in various crops, which may explain the reduction in grain damage and improved field stability [26,34].

Regarding grain yield, a noticeable effect of positive heterosis was identified in directed crosses between pigmented genotypes. This behavior is attributed to the genetic complementarity between selected parents, enhancing processes such as resource use efficiency, physiological stability under environmental variability, and effective allocation of photoassimilates toward storage structures [35,36]. The results also indicate that achieving sustained yield improvement requires identifying parental combinations that contribute not only complementary morphophysiological traits but also stability in key characteristics such as grain health, vegetative vigor, and functional metabolite expression [37,38].

In this regard, the incorporation of pigmented native lines into genetic improvement schemes may offer significant agronomic benefits. In particular, directed crosses involving LLMJ# and JCTM# showed a favorable combination of functional traits—such as stable anthocyanin expression—and superior grain yield, suggesting a positive heterotic effect on productivity.

### 3.3. Socioeconomic Relevance and Sustainability

Beyond its visual appeal, this trait offers functional and agri-food advantages. Purple totomoxtle, traditionally used for making tamales, acquires added value as a potential source of bioactive compounds [39]. Recent studies highlight that pigmented plant residues can be transformed into natural colorants, functional antioxidants, or feed additives for ruminants—reducing methane emissions and improving digestibility [26,40]. In addition to its cultural relevance as a traditional wrapping, totomoxtle could represent an underutilized source of bioactive compounds, such as anthocyanins and flavonoids. Future studies should focus on characterizing its nutritional and antioxidant profile, as well as evaluating its sensory properties and its potential for industrial applications, including food, pharmaceutical, and cosmetic products.

These findings support the development of value-added chains based on circular economy principles.

The results of this study position pigmented native maize not only as a component of biocultural heritage but also as a strategic resource for developing sustainable, resilient agricultural models with high added value—particularly in rural communities where maize is central to food security, livelihoods, and cultural identity [25,41].

## 4. Materials and Methods

### 4.1. Location of Experimental and Analytical Phases

This study was conducted across three specialized facilities that integrated agronomic evaluation and biochemical analysis phases. Activities related to conservation, phenotypic selection, directed crosses, and field trial establishment were carried out at the San Ramón University Center for Technological Transfer (CUTT), part of the Universidad Autónoma de Chiapas (UNACH), located in the municipality of Villaflores, Chiapas.

Photosynthetic pigment analyses (chlorophylls and carotenoids) were performed at the Plant Tissue Culture Laboratory of the Tecnológico Nacional de México, Tuxtla Gutiérrez Campus, while anthocyanin quantification was conducted at the Biotechnology Laboratory of the Instituto de Ciencias Agrícolas (ICA) at the Universidad Autónoma de Baja California (UABC).

### 4.2. Plant Material and Experimental Conditions

Since 2000, a participatory conservation and improvement program for native maize has been implemented in the Frailesca region of Chiapas, Mexico. The program began with the collection of approximately 1 kg of seed per accession, gathered directly from local farmers’ fields. This germplasm has been preserved and evaluated annually under rainfed conditions in experimental plots composed of eight 10 m long rows. Selection of materials has been based on phenotypic criteria established by the local custodians of the germplasm, prioritizing traits such as compact ear coverage, well-structured plants, and low ear insertion height.

From 2010 to 2022, the program continued with activities focused on maintenance, phenotypic selection, and genetic purification of the germplasm, leading to the preservation of key agronomic and morphological traits. In 2022, twenty-four populations were evaluated, of which fourteen were selected and conserved based on the outstanding performance of specific family lines. Subsequently, in 2023, manual pollinations were conducted to maintain genetic identity and generate inter-parental crosses. These activities took place at the San Ramón University Center for Technological Transfer (CUTT) of the Universidad Autónoma de Chiapas (UNACH), with the goal of developing advanced lines that would stably express the purple totomoxtle phenotype—a trait of functional, cultural, and culinary significance.

This characteristic is illustrated in Figure 5, which shows maize plants with visible purple pigmentation at harvest (A), as well as the traditional use of pigmented totomoxtle in Mexican cuisine, specifically in the preparation of tamales wrapped in purple husks (B).

During the 2024 spring–summer cycle, an experimental trial was established with ten genotypes, including pigmented parental lines from fraternal and inter-parental crosses, along with control materials with and without anthocyanin content. Table 4 details the genealogy, origin, and phenotypic pigmentation traits of each evaluated material, complemented by genetic and agronomic information previously reported by Llaven et al. [23].

This conservation and improvement scheme has enabled the preservation of traditional high-value traits, such as purple pigmentation in the totomoxtle, leaf sheath, and stem, features associated with nutraceutical properties and potential mechanisms of pest resistance and tolerance to abiotic stress factors.

### 4.3. Growing Conditions and Agronomic Management

The experiment was conducted under rainfed conditions at the CUTT San Ramón facility (Villaflores, Chiapas), on deep soils with a loam–clay–sandy texture and an average annual precipitation of approximately 1222 mm. However, the 2024 spring–summer cycle experienced a prolonged drought, which affected vegetative development and increased biotic pressure.

The experimental area was selected for its homogeneous soil and topography. The plots were evenly distributed, and all received the same natural rainfall, with no irrigation applied. Land preparation was carried out using two perpendicular passes with a disc harrow, enabling initial mechanical weed control. Manual sowing was performed on 23 June 2024, using three rows per treatment, with two full replicates of the trial: the first replicate was designated for biochemical sampling and the second for grain yield evaluation. A completely randomized design was used, and three internal repetitions per row were implemented to control for environmental and genotypic variability in the native germplasm.

Due to high initial weed pressure—typical of the humid tropics—a pre-emergent treatment was applied five days after sowing (DAS), consisting of atrazine (1 kg ha^−1^) combined with paraquat + diuron (2 L ha^−1^), aimed at minimizing early-stage competition for resources. Two additional weed control applications followed: the first at 54 DAS using the same formulation, and the second at 80 DAS, applied only to row pathways to maintain clean coverage and enhance airflow at the plant base during the reproductive stage.

Thinning was conducted at 14 DAS, leaving two plants per planting point, resulting in a final density of 64,100 plants ha^−1^, with 0.4 m spacing between plants and 0.78 m between rows. This configuration ensured a uniform distribution and balanced crop development.

Fertilization was applied in two strategic phases. The first occurred at 18 DAS, with a mixture of 200 kg ha^−1^ of diammonium phosphate (DAP) and 100 kg ha^−1^ of urea, placed manually 5 cm from the base of each plant. The second fertilization was performed at 47 DAS, with an application of 350 kg ha^−1^ of urea to meet nutritional demands during stem elongation and reproductive structure formation.

As part of pest management, two insecticide applications were carried out to control fall armyworm (*Spodoptera frugiperda*), a pest that typically shows low incidence in native maize but presented significant pressure during this drought-affected cycle. The first application was at 20 DAS with emamectin (150 mL ha^−1^) as a preventive measure during early development. The second was at 48 DAS with cypermethrin (250 mL ha^−1^), in response to visible damage in the central whorl, particularly in anthocyanin-deficient plots, which showed greater susceptibility.

No fungicide treatments were applied, as no relevant disease symptoms were observed in any of the plots—likely due to the inherent vigor of the native materials. At 135 DAS, with plants ten days past physiological maturity, dry totomoxtle samples were collected for anthocyanin analysis. Full harvest took place at 156 DAS, and grain yield was quantified per row for each plot.

### 4.4. Evaluated Agronomic Variables

The variables assessed in this study were classified into three main categories: agronomic performance, ear quality and health indicators, and visual presence of anthocyanins in vegetative structures. Grain yield (t ha^−1^) was calculated based on the total grain weight per plot, adjusted to a commercial moisture content of 14%, considering the number of plants and ears harvested per experimental unit. These variables were recorded at harvest for each treatment established in the field.

For quality indicators, overall ear appearance was evaluated through direct visual inspection at harvest, considering uniformity in size, consistency among ears within each plot, and absence of external damage. A score of 9.0 was assigned to ears with the best phenotypic characteristics (compact, undamaged, and well-formed), with descending values given to ears of lower visual quality. This methodology was adapted jointly with germplasm custodians since 2000, based on phenotypic selection criteria used in the native germplasm conservation program.

After shelling, grain health was assessed using an ordinal scale with three categories: 1 (healthy), 2 (intermediate, with partial damage), and 3 (damaged, due to fungi, insects, or visible rot). Additionally, the total biomass of damaged grain per plot was quantified in grams as a complementary indicator of the sanitary condition of each genotype.

The presence and visual intensity of anthocyanins in vegetative structures were evaluated at two key stages of the crop cycle. The first assessment was conducted at 43 days after sowing (DAS), during the vegetative stage, recording the number of pigmented nodes and the intensity of pigmentation. A second evaluation was conducted at 65 DAS (n = 25), focusing on the intensity and distribution of anthocyanin pigmentation in stems and basal sheaths. Plants were classified according to the visual intensity of pigmentation in these structures.

The second major evaluation was performed at harvest through the analysis of totomoxtle in transverse sections of maize ears. In both assessments, a visual scale based on the methodology described in the *Manual gráfico para la descripción varietal en maíz* [42] was used, categorizing pigment intensity into four levels: absent or very weak, weak, moderate, and strong. The classification scale used is shown in Figure 6.

Additionally, for treatments that exhibited anthocyanin expression, representative photographs of vegetative and reproductive organs were taken to visually document the pigment accumulation potential in multiple structures of these genotypes.

### 4.5. Total Anthocyanin Content

Total anthocyanin content was determined using the differential pH method described by Jungmin et al. [43], with modifications. Sampling days were selected based on key phenological stages related to pigment expression and the plant’s physiological activity. For leaf sheaths, samples were collected on days 19, 43, and 65, corresponding to early vegetative growth, stem elongation, and pre-tassel emergence, respectively—critical stages for assessing pigment biosynthesis during active development (Figure 7A–C). For totomoxtle, samples were taken on days 72 and 135, corresponding to the milk stage (when ears are harvested tender for consumption) and physiological maturity (when dry husks are used for culinary purposes and, in some cases, as forage) (Figure 7D).

Extracts were prepared using 5 g of plant material in an ethanol–distilled water solution (2:3 *v*/*v*). Two tubes were prepared for each extract (labeled A and B), to which 500 µL of extract and 500 µL of ethanol containing 0.01% HCl (*v v*^−1^) were added. Then, 5 mL of solution A (aqueous HCl 2% *v v*^−1^) was added to tube A, and 5 mL of solution B (a pH 3.5 buffer containing 0.2 M Na_2_HPO_4_ and 0.1 M citric acid) was added to tube B.

Absorbance was measured at 520 nm using a UV-Vis spectrophotometer (Model DR6000™ UV-Vis, Hach, Loveland, CO, USA). Total anthocyanin content was calculated using the following equation:Total anthocyanins (mg/L) = (A_1_ − A_2_) × ff = 396.598A_1_ = Absorbance of tube AA_2_ = Absorbance of tube B

### 4.6. Photosynthetic Pigment Quantification

To quantify photosynthetic pigments (chlorophylls and carotenoids), 50 mg of fresh leaf tissue (Figure 6A–C) was used for extraction with 1.5 mL of 80% acetone (Sigma-Aldrich^®^, St. Louis, MO, USA). The samples were incubated in darkness at 4 °C for one hour to maintain pigment stability, followed by centrifugation at 10,000 rpm for 5 min. The procedure was based on the methodologies described by Inskeep and Bloom [44] and Biehler et al. [45]. The absorbance of the supernatant was measured using a UV-Vis spectrophotometer (Model DR6000™ UV-Vis, Hach, Loveland, CO, USA) at specific wavelengths: 664 nm for chlorophyll a (Chl a), 647 nm for chlorophyll b (Chl b), and 450 nm for carotenoids. Pigment concentrations were calculated using the following equations:Chl a (µg mL^−1^) = 12.7 × A664 − 2.79 × A647Chl b (µg mL^−1^) = 20.7 × A647 − 4.62 × A664Total chlorophyll = Chl a + Chl bCarotenoids (µg mL^−1^) = (A_MAX_ × M × 1000)/(ε × δ)
where M is the molecular weight of β-carotene (537 g mol^−1^), ε is the molar extinction coefficient in acetone (140,663 L mol^−1^ cm^−1^), and δ is the path length (cm).

### 4.7. Experimental Design and Statistical Analysis

The field experiment was established under a completely randomized design (CRD), using ten native maize genotypes as treatments, with three replicates, each consisting of three 5 m long rows. To ensure data reliability and allow independent management of agronomic and biochemical variables, two parallel replicates of the experiment were implemented: Replicate 1 was designated for agronomic and yield evaluation, while Replicate 2 was used exclusively for laboratory sampling (biochemical analyses).

All data were subjected to analysis of variance (ANOVA), and mean comparisons were performed using Tukey’s test (*p* ≤ 0.05), using SAS software version 9.4. Prior to ANOVA, assumptions of normality and homogeneity of variances were verified.

The evaluation of pigments (anthocyanins in leaf sheaths and totomoxtle, as well as photosynthetic pigments: chlorophyll a, chlorophyll b, total chlorophyll, and carotenoids) was carried out cumulatively by treatment, using a randomized complete block design (RCBD), with sampling days as blocks (19, 43, and 65 days after sowing for leaf sheaths and photosynthetic pigments; 72 and 135 days for totomoxtle) and genotypes as treatments. This approach allowed for the identification of the phenological stage with the highest pigment content and the assessment of its temporal dynamics.

The visual expression of anthocyanins in stems (43 DAS) and the pigmentation classification of whole plants (65 DAS) were evaluated under a completely randomized design (CRD) with three replicates per treatment, categorizing visible intensity as high, medium, or absent.

Finally, for agronomic variables (overall ear appearance, damaged grain biomass, grain health, and grain yield), a CRD was used with three replications per treatment.

## 5. Conclusions

This study demonstrated the effectiveness of participatory breeding and phenotypic selection in conserving and enhancing the purple totomoxtle trait in native maize from the Frailesca region of Chiapas. Directed crosses between pigmented parent lines promoted both the intensification of anthocyanin accumulation in vegetative and reproductive structures, as well as increased grain yield.

Genotypes with outstanding agronomic traits were identified, particularly regarding ear quality, grain health, and functional accumulation of photosynthetic pigments. The diversity expressed among the evaluated materials supports their potential as a genetic base for breeding programs with an agroecological focus.

The presence of anthocyanins in vegetative structures from early growth stages, along with the expression of purple totomoxtle, in addition to its cultural value, may serve as a useful visual biomarker for selection strategies, agrobiodiversity conservation, and the strengthening of traditional agricultural systems.

## Figures and Tables

**Figure 1 plants-14-02511-f001:**
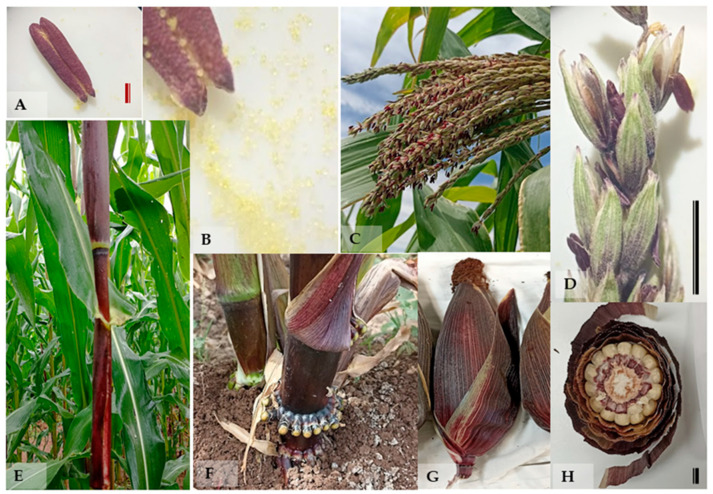
Structural distribution of anthocyanins in native maize. (**A**) Individual anther with visible pigmentation; scale bar: 1 mm. (**B**) Pollen release from a pigmented anther. (**C**) Male inflorescence during anthesis (pollen release). (**D**) Single spikelet with anther in dehiscence and pollen discharge; scale bar: 10 mm. (**E**) Pigmentation in the stem and leaf sheath. (**F**) Basal nodes and adventitious roots with visible anthocyanin pigmentation. (**G**) Ear with purple-colored husk and stigmas. (**H**) Transverse section of a mature ear showing the purple bracts (totomoxtle), the cob (rachis), and white kernels; scale bar: 10 mm.

**Figure 2 plants-14-02511-f002:**
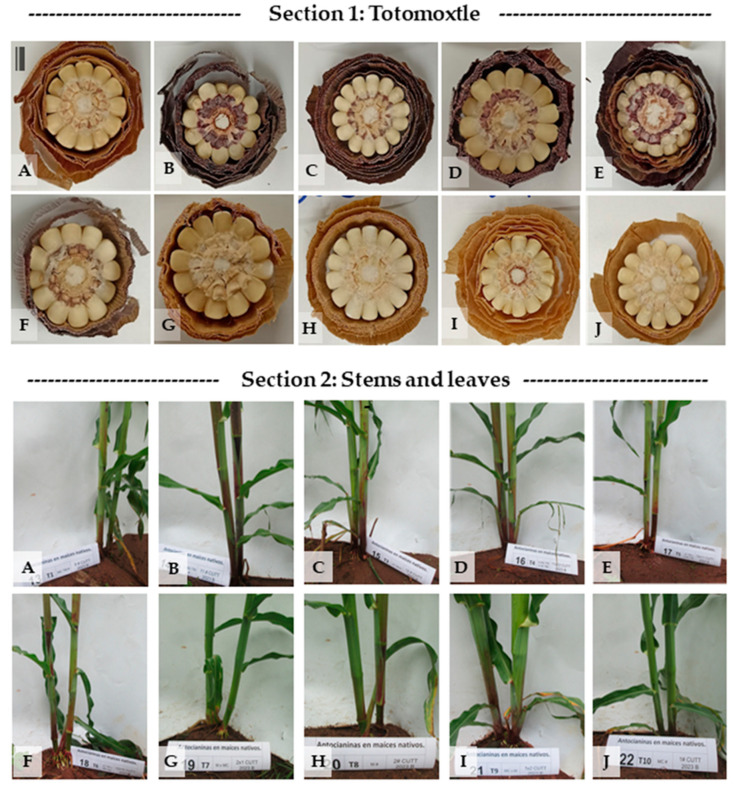
Anthocyanin pigmentation variability in 10 native maize genotypes. Section 1: Totomoxtle (transverse sections of ears at harvest; totomoxtle refers to the pigmented ear bracts surrounding the kernels). Section 2: Stems and leaf sheaths at basal nodes, evaluated on day 43. (**A**) = MCTM#: Sibling cross of a control line with purple totomoxtle (TM). (**B**) = LLMJ#: Selected progenitor with totomoxtle pigmentation. (**C**) = JCTM#: Selected progenitor with totomoxtle pigmentation. (**D**) = LLMJ × JCTM: Direct inter-population fraternal cross between (**B**,**C**). (**E**) = JCTM × LLMJ: Fraternal reciprocal cross between (**C**,**B**). (**F**) = JCTM × RP Rojo: Cross with a female progenitor with pigmented totomoxtle (JCTM) and a male with anthocyanin pigmentation in the grain. (**G**)= Mor × MC: Reciprocal cross (**H**) × (**J**). (**H**) = Mor#: Unselected progenitor 2. (**I**) = MC × Mor: Reciprocal cross (**J**) × (**H**). (**J**) = MC#: Unselected progenitor 1. (**G**–**J**): Control native maize genotypes not selected for anthocyanin pigmentation. Images (**A**–**J**) correspond to both analyzed structures: totomoxtle (Section 1) and stem/sheath (Section 2).

**Figure 3 plants-14-02511-f003:**
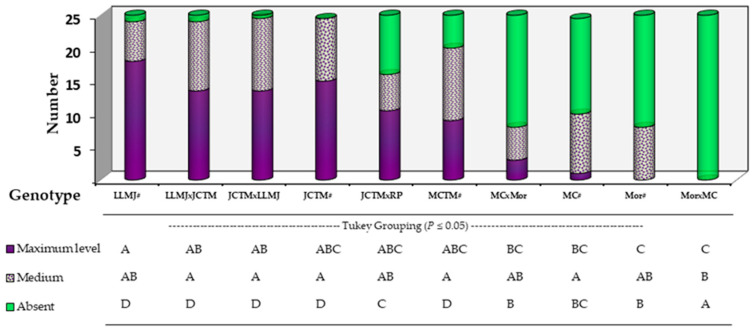
Plant classification by visible anthocyanin intensity across whole vegetative structures in native maize genotypes (day 65, BBCH 67). Different letters under each bar, within each factor and level, indicate significant differences (*p* ≤ 0.05).

**Figure 4 plants-14-02511-f004:**
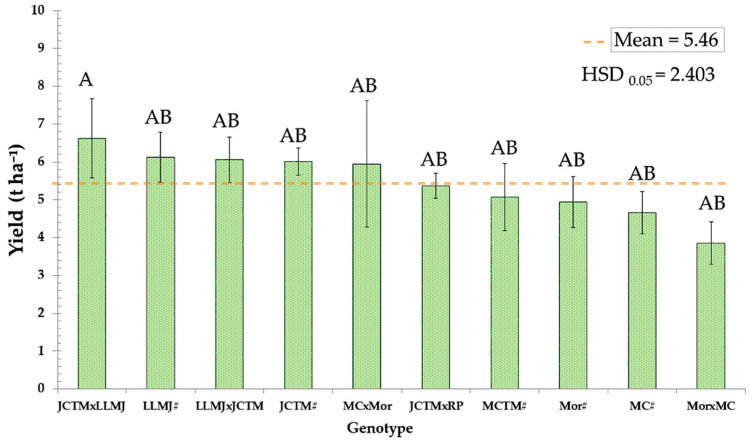
Grain yield of sibling-cross parents and their inter-population crosses of native maize according to anthocyanin presence. Different letters under each bar, within each factor, indicate significant differences (*p* ≤ 0.05).

**Figure 5 plants-14-02511-f005:**
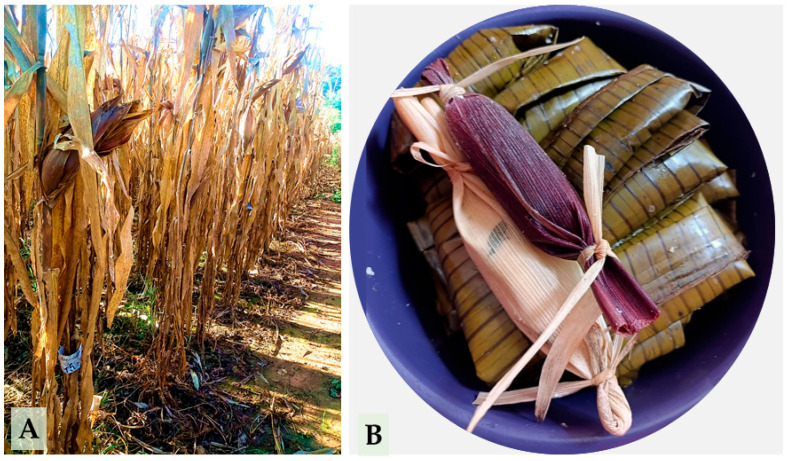
Purple totomoxtle in native maize: visual expression in the field and application in traditional gastronomy. (**A**) Presence of pigmentation in plants at the time of harvest. (**B**) Tamales wrapped with pigmented totomoxtle husks.

**Figure 6 plants-14-02511-f006:**
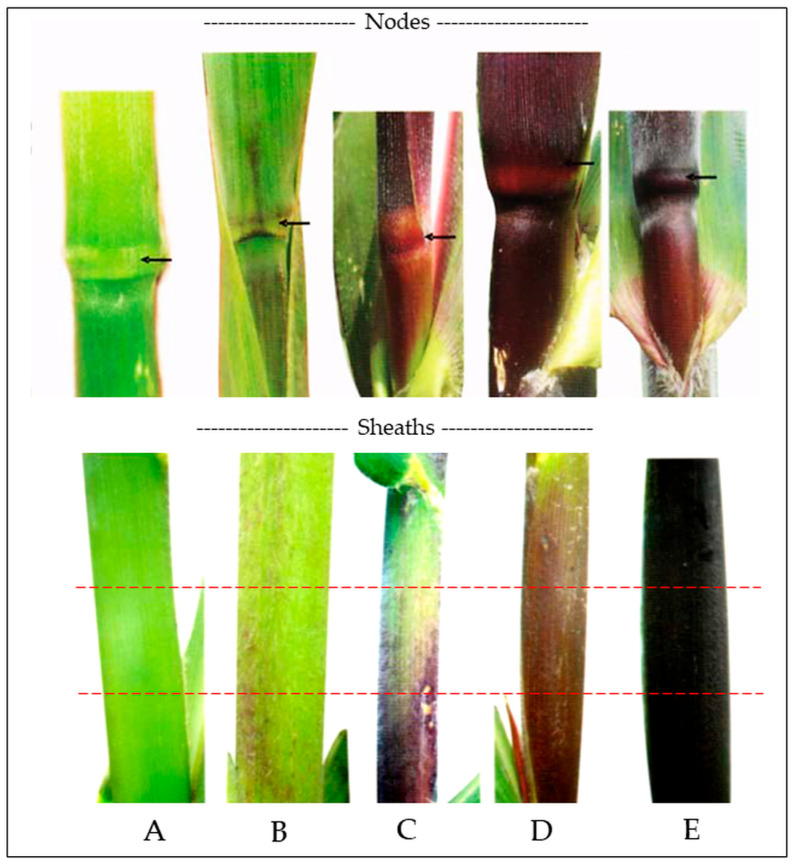
Anthocyanin pigmentation in nodes and leaf sheaths (red lines indicate sampling zones). (**A**) = absent or very weak, (**B**) = weak, (**C**) = moderate, (**D**) = strong, (**E**) = very strong. Adapted from Carballo and Ramírez, [42].

**Figure 7 plants-14-02511-f007:**
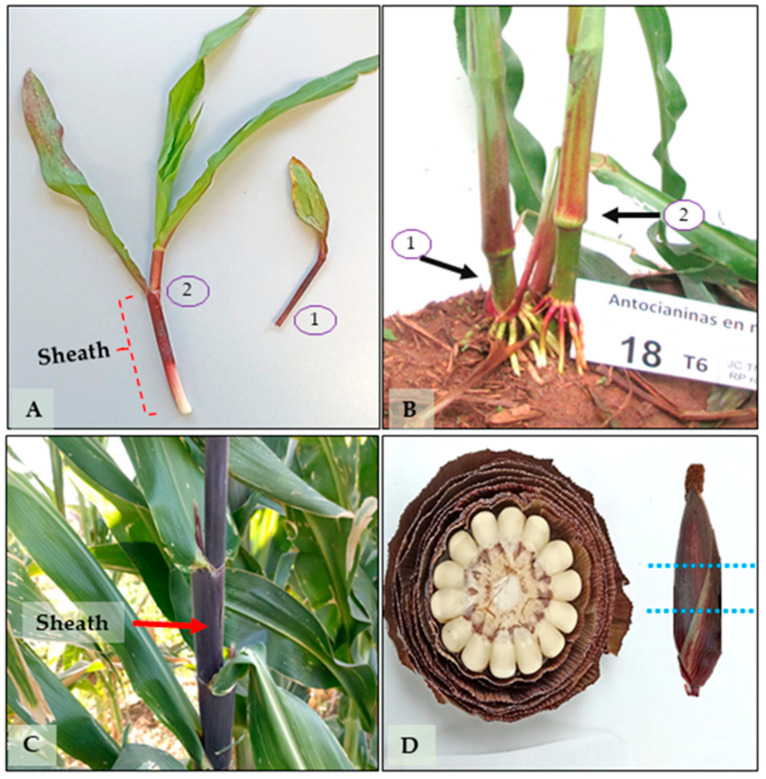
Plant structures selected for anthocyanin analysis during five phenological stages in native maize. (**A**) Seedling at day 19 showing leaf 1, leaf 2, and the sheath (indicated with red lines). (**B**) Plant at day 43 with two selected sheaths indicated, based on the position of the adventitious roots. (**C**) Plant at day 65 showing the position of the first silking and the sampled sheath. (**D**) Transverse section of the ear showing the totomoxtle sampling zones (blue lines) used for anthocyanin extraction, corresponding to day 72 (milky grain stage) and day 135 (dry ear stage).

**Table 1 plants-14-02511-t001:** Presence and intensity of anthocyanins in stem nodes at 43 days in sibling-cross parents and their inter-population crosses of native maize.

Genealogy	Number/Nodes	Intensity
LLMJ × JCTM	7.0 ^A^	6.0 ^AB^
LLMJ#	7.0 ^A^	8.0 ^A^
JCTM × LLMJ	7.0 ^A^	8.0 ^A^
JCTM#	6.0 ^AB^	5.0 ^ABC^
JCTM × RP	5.5 ^AB^	4.0 ^ABC^
MC × Mor	4.9 ^AB^	3.0 ^BC^
MCTM#	4.5 ^AB^	3.0 ^BC^
MC#	4.2 ^B^	2.0 ^BC^
Mor#	4.0 ^BC^	2.0 ^BC^
MorxMC	1.5 ^C^	1.0 ^C^
DMS	2.54	4.34

The means followed by a different letter within column are significantly different at *p* < 0.05.

**Table 2 plants-14-02511-t002:** Dynamics of anthocyanin content in leaf sheath and totomoxtle during the phenological cycle of sibling-cross parents and their inter-population crosses of native maize.

Pedigree	Sheath (Leaf)	Totomoxtle
	mg L^−1^	
LLMJ#	142.3 ^A^	376.9 ^A^
JCTM#	100.8 ^AB^	233.8 ^B^
JCTM × LLMJ	94.5 ^AB^	223.8 ^B^
LLMJ × JCTM	72.9 ^BC^	174.6 ^B^
MCTM#	66.2^BC^	93.8 ^C^
MC × Mor	65.7 ^BC^	40.3 ^C^
JCTM × RP	59.6^BC^	65.5 ^C^
Mor#	35.9 ^C^	25.5 ^C^
MC#	24.0 ^C^	37.5 ^C^
Mor × MC	23.9 ^C^	22.7 ^C^
*p*	0.0001	
r^2^	0.9	
Day 19	79.38 ^A^	*
Day 43	57.559 ^A^	*
Day 65	68.80 ^A^	*
Day 72	*	95.27 ^B^
Day 135	*	123.58 ^A^
*p*	0.072	0.0102

The means followed by a different letter within column are significantly different at *p* < 0.05.

**Table 3 plants-14-02511-t003:** Indicators of ear quality and health as agroecological selection criteria for sibling-cross parents and their inter-population crosses in native maize genotypes.

Pedigree	General Appearance	Damaged Grain Biomass (g)	Grain Health
JCTM × LLMJ	8.400 ^ABCD^	0.208 ^A^	1.00 ^A^
LLMJ#	8.667 ^AB^	0.410 ^A^	1.33 ^A^
LLMJ × JCTM	8.267 ^CD^	0.349 ^A^	1.00 ^A^
JCTM#	8.700 ^A^	0.149 ^A^	1.00 ^A^
MC × Mor	8.467 ^ABC^	0.177 ^A^	1.33 ^A^
JCTM × RP	8.500 ^ABC^	0.234 ^A^	1.00 ^A^
MCTM#	8.333 ^BCD^	0.089 ^A^	1.33 ^A^
Mor#	8.067 ^D^	0.238 ^A^	1.33 ^A^
MC#	8.367 ^ABCD^	0.295 ^A^	1.17 ^A^
Mor × MC	8.333 ^BCD^	0.367 ^A^	1.67 ^A^
Average	8.410	0.252	1.22
HSD = 0.05	0.346	0.529	0.9143

The means followed by a different letter within column are significantly different at *p* < 0.05.

**Table 4 plants-14-02511-t004:** Native maize genotypes evaluated during the spring–summer 2024 cycle and characteristics of their genetic origin.

Genealogy	Local Name	Origin	Observation	Notes
Anthocyanin-positive control				
MC TM #	Macho TM-#	9 # CUTT 2023 B	Progenitor 9	Selection TM (#) PV 2023
Populations after separating anthocyanin-containing families				
LLMJ #	Llamaja TM-#	11 # CUTT 2023 B	Progenitor 11	Selection TM (#) PV 2023
JC TM #	Jaime Camacho TM-#	12 # CUTT 2023 B	Progenitor 12	Selection TM (#) PV 2023
LLMJ × JC TM	Llamaja TM-# × Jaime Camacho TM-#	11 × 12 CUTT 2023 B	Offspring (♀11 × ♂12)	Direct cross TM PV 2023
JC TM x LLMJ	Jaime Camacho TM-# × Llamaja TM-#	12 × 11 CUTT 2023 B	Offspring (♀12 × ♂11)	Reciprocal cross TM PV 2023
Control with anthocyanins in the female progenitor				
JC TM x RP Rojo	Jaime Camacho TM-# × Rojo Parral	12 × 10 CUTT 2023 B	Offspring (♀12 × ♂10)	Cross TM × Red grain PV 2023
----- Controls without selection for anthocyanins/genetic purity maintenance cycle -----				
M × MC	Morales × Macho	2 × 1 CUTT 2023 B	Offspring (♀2 × ♂1)	Direct cross between native varieties
M #	Morales #	2# CUTT 2023 B	Progenitor 2	Original control (#) PV 2023
MC × M	Macho × Morales	1 × 2 CUTT 2023 B	Offspring (♀1 × ♂2)	Reciprocal cross between native varieties
MC #	Macho #	1# CUTT 2023 B	Progenitor 1	Original control (#) PV 2023.

TM = Purple totomoxtle. # = Sibling cross. “TM selection (#)” refers to materials derived from visually selected pigmented plants, separated by families during the PV 2023 cycle. “Direct cross” and “Reciprocal cross” refer to the type of hybridization performed between native genotypes with or without pigmentation. “Original control” corresponds to base materials preserved at CUTT San Ramón.

## Data Availability

The data presented in this study are available on request from the corresponding author.

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
