# Peer review of "Variability in Anthocyanin Expression in Native Maize: Purple Totomoxtle as a Phenotypic Trait of Agroecological Value"

_plants, 2025, doi:10.3390/plants14162511_

Round 1

Reviewer 1 Report

Comments and Suggestions for Authors

The manuscript illustrates a well-structured and insightful investigation into the phenotypic diversity of anthocyanin expression in native maize varieties, especially focusing on the purple totomoxtle trait for sustainable agriculture. The study is commendable for linking biochemical and phenotypic characterization with agroecological relevance, particularly in the context of local biodiversity and traditional farming systems. The experimental design is sound, and the data interpretation is generally credible. However, the following points should be taken into consideration:

Abstract:

The author writes the abstract based on their obtained findings.

Introduction: The author precisely addresses the research gap with proper justification. However, the author needs to address the previous research findings concisely and clearly.

Results:

In Tables 2 and 3: How do you get the data from 19, 43, and 65 days? Are these cumulative of the studied lines (10 lines)? Please explain.

Discussion:

Line 358-401: Please write in English, not in another language.

The author discussed their obtained findings with proper justification.

Materials and Methods:

Line 626-630: The author mentioned that leaf sheath samples were collected on 19, 43, and 65 days (Figure 6A–C), and totomoxtle samples were collected on days 72 and 135 (Figure 6D). What about Figure 6E? Please rewrite with appropriate sentence structure. For example, 19, 43, and 65 days, respectively.  

Is the data you collected cumulative from all cultivars/lines or a single line? Please mention clearly.

Conclusion:

The author needs to write this section concisely and mention only the key findings.

References:

The author needs to check the reference lists that are cited in the text. Needs to write every scientific name in Italics. The title of the cited reference should be in English.

Reviewer 2 Report

Comments and Suggestions for Authors

Dear Authors,

I have reviewed the manuscript. This is an original and timely manuscript that investigates anthocyanin pigmentation in native maize genotypes, with particular focus on the purple totomoxtle trait. The study is relevant to agroecological breeding and the preservation of cultural heritage, and it provides new quantitative data on pigmentation-related traits.

The introduction is thorough and well-contextualized, although it would benefit from a clearer distinction between previously established findings and the specific objectives of the current research.

The results are extensive and well-documented, supported by informative tables and figures. However, the visual overload could be reduced by merging some tables or moving parts of them to the supplementary materials. The discussion effectively integrates agronomic, biochemical, and cultural aspects, but the mechanistic interpretations (e.g., maternal effects, epistasis) would require at least brief molecular references or more cautious phrasing.

Reviewer 3 Report

Comments and Suggestions for Authors

I suggest:

Abstract: Suggest adding one sentence on practical implications (e.g., breeding targets).

Introduction: Add 1–2 sentences to clarify the hypothesis or research questions explicitly.

While the differential pH method and photosynthetic pigment protocols are adequately cited, a clearer rationale for the selected time points (days 19, 43, 65, 72, 135) would be helpful.

It’s unclear how field variability (e.g., soil heterogeneity, rainfall distribution) was controlled or measured, especially considering the drought conditions mentioned.

The experiment was conducted in one location (Chiapas) over one growing season. This limits generalizability across environments or seasons.

The labeling system (e.g., LLMJ#, JCTM#) may confuse readers unfamiliar with the breeding program. Include a summary table or pedigree diagram showing parentage, pigmentation phenotype, and key traits (yield, pigment levels, etc.)..

Although chlorophyll and carotenoid levels are measured, the discussion of their role in stress response or photosynthetic efficiency is minimal. Expand the discussion on how pigmentation might be physiologically or adaptively linked to abiotic stress (e.g., drought tolerance).

The discussion of the potential link between anthocyanins and stress resilience is promising but could benefit from referencing more current molecular or transcriptomic studies.

Although inheritance is discussed, no molecular or genetic marker analysis is provided to support claims about additive or epistatic effects.

Environmental variability (e.g., drought) is acknowledged but not statistically controlled for or correlated with pigment data.

While cultural uses (e.g., tamales) are described, the totomoxtle is not analyzed for nutritional, antioxidant, or industrial potential. Add 2–3 sentences in the Discussion suggesting future biochemical or sensory analysis of totomoxtle.

Several tables are densely packed with data. Consider splitting or highlighting key comparisons (e.g., top 3 yielding genotypes).

Some tables (e.g., Table 3) could be moved to supplementary materials or presented more concisely.

Add a summary figure or table showing the genotype-by-trait matrix (pigment level, yield, grain health, etc.) to guide interpretation.

Round 2

Reviewer 3 Report

Comments and Suggestions for Authors

The authors responded point by point The manuscript appears to have been revised in good faith and with attention to scientific rigor and clarity.